# Advances in NAD-Lowering Agents for Cancer Treatment

**DOI:** 10.3390/nu13051665

**Published:** 2021-05-14

**Authors:** Moustafa S. Ghanem, Fiammetta Monacelli, Alessio Nencioni

**Affiliations:** 1Department of Internal Medicine and Medical Specialties (DIMI), University of Genoa, Viale Benedetto XV 6, 16132 Genoa, Italy; Moustafa.Ghanem@edu.unige.it (M.S.G.); fiammetta.monacelli@unige.it (F.M.); 2Ospedale Policlinico San Martino IRCCS, Largo Rosanna Benzi 10, 16132 Genova, Italy

**Keywords:** NAD, cancer, metabolism, NAMPT inhibitors, NAPRT, salvage pathway, Preiss–Handler pathway, de novo pathway, vitamin B3

## Abstract

Nicotinamide adenine dinucleotide (NAD) is an essential redox cofactor, but it also acts as a substrate for NAD-consuming enzymes, regulating cellular events such as DNA repair and gene expression. Since such processes are fundamental to support cancer cell survival and proliferation, sustained NAD production is a hallmark of many types of neoplasms. Depleting intratumor NAD levels, mainly through interference with the NAD-biosynthetic machinery, has emerged as a promising anti-cancer strategy. NAD can be generated from tryptophan or nicotinic acid. In addition, the “salvage pathway” of NAD production, which uses nicotinamide, a byproduct of NAD degradation, as a substrate, is also widely active in mammalian cells and appears to be highly exploited by a subset of human cancers. In fact, research has mainly focused on inhibiting the key enzyme of the latter NAD production route, nicotinamide phosphoribosyltransferase (NAMPT), leading to the identification of numerous inhibitors, including FK866 and CHS-828. Unfortunately, the clinical activity of these agents proved limited, suggesting that the approaches for targeting NAD production in tumors need to be refined. In this contribution, we highlight the recent advancements in this field, including an overview of the NAD-lowering compounds that have been reported so far and the related in vitro and in vivo studies. We also describe the key NAD-producing pathways and their regulation in cancer cells. Finally, we summarize the approaches that have been explored to optimize the therapeutic response to NAMPT inhibitors in cancer.

## 1. Introduction

Cancer cells share common distinctive features that dictate their aberrant behavior in the body. Hanahan and Weinberg described six capabilities that tumor cells acquire which enable their growth and proliferation, and named them “the hallmarks of cancer”. These hallmark features consist in the ability of malignant cells to sustain the self-supply of growth signals, escape anti-growth signals, evade programmed cell death, maintain the formation of new blood vessels, possess an infinite replicative potential, and, finally, to invade and metastasize into distant tissues [1]. More than a decade later, the same authors added additional features as emerging hallmarks of cancer. The reprogramming of cellular metabolism is among these “next-generation” hallmarks that ultimately support malignant cell survival [2].

Among the major reprogrammed metabolic processes in cancer cells, there is what is named the Warburg’s effect (or aerobic glycolysis) after the name of its discoverer [3]. Typically, in non-malignant cells, in the presence of oxygen, glucose undergoes glycolysis to produce pyruvate which is then converted to acetyl Co-A and enters the citric acid cycle in the mitochondria to produce CO_2_ and energy in the form of adenosine triphosphate (ATP) via oxidative phosphorylation. However, in cancerous cells, glucose metabolism is rewired and, even in the presence of oxygen, glycolysis predominates while oxidative phosphorylation is reduced, leading to the fermentation of pyruvate to lactate [3]. This so-called Warburg effect is closely related to the need that cancer cells have to sustain a higher production of nicotinamide adenine dinucleotide (NAD), a central molecule required not only at the physiological level but also by neoplastic cells to support numerous crucial cellular processes [4].

NAD is a pyridine nucleotide essential coenzyme that plays a major role in the oxidation–reduction reactions taking place inside the cell, and is also responsible for energy production. Through behaving as an electron acceptor/donor that shuttles between oxidized and reduced forms, NAD supports various metabolic pathways such as glycolysis, Krebs cycle (citric acid cycle), oxidative phosphorylation, and fatty acid oxidation [5,6]. Beyond its role in bioenergetics and cellular metabolism, NAD possesses a prominent cell regulatory function. NAD serves as a substrate for NAD-dependent enzymes such as mono (ADP-ribosyl) transferases (MARTs), poly (ADP-ribose) polymerases (PARPs), sirtuins (SIRT1-7), and cyclic ADP-ribose (cADPR) synthases (CD38 and CD157) [7,8,9]. Indeed, NAD is degraded during the poly-ADP ribosylation reactions carried out by PARPs (e.g., PARP1/2), during protein deacetylation reactions through sirtuins (e.g., SIRT1), and by the hydrolase and ADP-ribosyl cyclase activities of CD38 and CD157 [8,9]. These NAD-consuming reactions orchestrate a series of fundamental biological processes including gene expression, transcription, calcium signaling, DNA repair, apoptosis, circadian rhythm, and cell cycle progression [8,10]. Importantly, many of these cellular events were found to be implicated in malignant transformation and cancer cell survival. For instance, cancer cells frequently face DNA-damaging insults and thus, typically rely on elevated PARP activity to carry out their DNA repair, overcome stress signals, and thereupon sustain their survival. As opposed to its fate as a coenzyme, NAD becomes rapidly degraded by the enzymatic reactions that use it as a substrate, potentially leading to a net NAD deficit [8]. As a consequence, a nonstop generation of NAD is required in normal tissues, and more in neoplastic cells, to compensate for the high NAD degradation. In light of the above findings, dysregulating NAD homeostasis through interfering with the NAD biosynthetic machinery and consequently reducing NAD pools within cancer cells has been conceived as a promising strategy for cancer treatment [8].

In this review, we provide an overview of NAD production routes in eukaryotic cells, the milestone findings, and studies in the development of NAD-depleting agents in the context of oncology therapeutics. In addition, the obstacles in the field and the most recent advances, including the ongoing clinical studies of NAD biosynthesis inhibitors, will also be highlighted.

## 2. NAD Biosynthesis in Mammals

Overall, mammalian cells utilize the following three biosynthetic pathways to generate NAD: the de novo pathway, the Preiss–Handler (PH) pathway, and the salvage pathway (Figure 1) [8,9]. In the de novo pathway, also referred to as the kynurenine pathway, the essential amino acid tryptophan serves as the starting molecule. It undergoes a cascade of enzymatic reactions that yield quinolinic acid (QA), which is then converted by the enzyme quinolinic acid phosphoribosyltransferase (QAPRT/QPRT) to nicotinic acid mononucleotide (NAMN) [11]. NAMN is also generated in the PH pathway by transferring a phosphoribosyl moiety from phosphoribosyl pyrophosphate (PRPP) to nicotinic acid (NA) with the help of another phosphoribosyltransferase enzyme called nicotinic acid phosphoribosyltransferase (NAPRT). NA is thus considered the precursor unit in the PH pathway of NAD synthesis [12,13]. Noticeably, PRPP, the provider of the NAD-sugar moiety, is synthesized from ribose-5-phosphate, which is a central product of the pentose phosphate pathway (PPP). In such a way, PRPP can be regarded as a linking molecule between glucose metabolism and NAD biosynthesis. The third biosynthetic pathway that contributes to maintaining the NAD supply is the salvage pathway, where NAD is generated starting from the end product of NAD-consuming enzymes, nicotinamide (NAM). Another phosphoribosyltransferase enzyme named nicotinamide phosphoribosyltransferase (NAMPT) controls the rate-limiting step in this pathway. NAMPT catalyzes the transfer of a phosphoribosyl group from the co-substrate PRPP to NAM, yielding nicotinamide mononucleotide (NMN) and pyrophosphate as a byproduct. Alternatively, the two mononucleotides NMN and NAMN can also be produced from the phosphorylation of the nucleosides nicotinamide riboside (NR) and nicotinic acid riboside (NAR) by nicotinamide riboside kinases (NMRK1/2) [14,15]. NAMN and NMN are converted to their corresponding dinucleotides nicotinic acid adenine dinucleotide (NAAD) and NAD through reactions governed by nicotinamide/nicotinic acid mononucleotide adenylyltransferases (NMNATs), of which three mammalian isoforms exist (NMNAT 1–3) [16,17]. In the final step of the PH pathway, NAD synthetase (NADSYN) catalyzes the amidation of NAAD into NAD using glutamine as a nitrogen donor.

NA, NAM and NR (collectively referred to as vitamin B3), as well as tryptophan, can all be obtained through diet, e.g., through cow milk in the case of NR [11,18]. As anticipated above, NAM is also largely produced “endogenously” in virtually every tissue as a byproduct of NAD-degrading enzymes, such as PARPs, sirtuins and CD38. NR can also be produced in bodily tissues starting from its reduced form, NRH, via NRH:quinone oxidoreductase 2 (NQO2), which utilizes NRH as an electron donor [19]. NRH was shown to be endogenously present in the liver by metabolomic analysis [20] and it was proposed to come from NADH degradation via NUDIX hydrolases, such as NUDIX5, 12, and 13 [21,22]. Notably, exogenously supplemented NRH can also stimulate NAD production independent of NMRK enzymes, through a mechanism that foresees its phosphorylation to NMNH, the reduced form of NMN, by adenosine kinase followed by the subsequent conversion to NADH and, ultimately, to NAD [20,23,24]. NRH and NMNH have indeed been identified as important NAD precursors both in vitro and in vivo [22,23,24].

Recent work by Shats et al. showed that gut bacteria convert NAM into NA through their nicotinamidase (PncA) and thereby contribute to NAD production in several mouse tissues, as well as in cancer cells growing the mouse, making them resistant to NAMPT inhibition (an effect that was shown to be dependent on NAPRT expression in cancer cells) [25]. These findings indicate that at least some amounts of gut microbiota- and diet-derived NA do reach bodily tissues despite the extensive metabolism NA undergoes in the liver (here, NA undergoes first-pass metabolism, being conjugated with glycine to form nicotinuric acid, which is then excreted in the urine, and being converted to nicotinamide—assumingly, via NA-mediated NAD generation and subsequent degradation with consequent NAM release). Indeed, circulating NA levels in the nanomolar range can be physiologically detected in mammals and such levels are increased by NA infusion [26] as well as by oral, pharmacological NA doses (gram doses of NA have been used for decades as a lipid-lowering approach). Thus, these observations suggest a role for dietary NA intake as well as for NA produced by the intestinal flora in fueling the PH pathway in NAPRT-expressing tissues (including NAPRT-positive tumors).

The role of NADH in the maintenance of the NAD pool should also not be overlooked. In this context, the enzyme NAD(P)H:quinone oxidoreductase (NQO1) oxidizes NADH into NAD, helping to maintain NAD levels [27]. Indeed, pharmacologically activating NQO1 was shown to augment the NAD/NADH ratio in the kidney tissues, meanwhile, *NQO1−/−* mice demonstrated higher NADH:NAD and NADPH:NADP ratios as compared to the wild-type mice [28,29].

Since many body tissues don’t express the complete set of the kynurenine pathway enzymes, the contribution of tryptophan to the overall NAD pool has long been thought to be less important. Nevertheless, a recent analysis of NAD synthesis and breakdown fluxes revealed that the liver mainly produces NAD from tryptophan, that it consumes NAD, producing NAM, and that the latter gets secreted in large amounts into the circulation to be used by other tissues [26]. This raises the interesting possibility that tryptophan might strongly contribute to the NAD pool of bodily tissues, not so much directly via de novo synthesis but rather indirectly through multistep inter-tissue cooperation. The latter involves (i) liver-dependent de novo synthesis, followed by (ii) the conversion of NAD into NAM (always in the liver), (iii) NAM excretion into the circulation, and (iv) finally, NAM uptake and utilization by the tissues to produce NAD via the canonical salvage pathway.

Recently published work proposes that the enzyme α-amino-β-carboxymuconate-ε-semialdehyde decarboxylase (ACMSD) acts as a metabolic gatekeeper of the de novo NAD biosynthetic route [30]. ACMSD is expressed mainly by the liver and kidney and it catalyzes the decarboxylation of α-amino-β-carboxymuconate-ε-semialdehyde (ACMS) to α-amino-β-muconate-ε-semialdehyde (AMS), thus rerouting tryptophan metabolism to picolinic acid and acetyl Co-A synthesis rather than towards NAD production [5,30]. Indeed, enhancement of de novo NAD synthesis and SIRT1 activity was observed in response to ACMSD inhibition [30]. In line with these findings, another study reported that mice overexpressing ACMSD show reduced blood and tissue NAD levels, as compared to their counterparts with normal ACMSD expression, when both groups were fed with niacin-free diets [31].

Finally, nicotinamide N-methyltransferase (NNMT) is another important modulating enzyme of NAD homeostasis. NNMT catalyzes the methyl transfer from S-adenosyl methionine (SAM) to NAM, yielding 1-methylnicotinamide, which is subsequently metabolized and excreted in urine [10]. NNMT derails NAM away from being recycled into NAD and curbs the NAM accumulation and inhibition of NAD-dependent signaling pathways by excessive NAM [32]. NNMT was found to regulate histone methylation and NAD-dependent SIRT1 signaling in adipose tissue and NNMT knockout was associated with elevated *NAMPT*, *NMNAT2* expression, and NAD levels in adipocytes, but not in the liver [33]. Notably, numerous cancer types and stromal cells upregulate NNMT expression, which epigenetically reprograms gene expression and causes a state of histone hypomethylation by consuming SAM methyl groups which, in turn, depletes SAM and creates a methyl sink [34,35]. Similarly, NNMT was shown to promote DNA hypomethylation in mesenchymal glioblastoma stems cells (GSCs) and to drive alterations in DNA methylation within the promoter regions of several genes in cancer-associated fibroblasts [35,36].

## 3. Regulation of NAD Production in Cancer Cells

The molecular basis and genetic mechanisms that underly the selection of the NAD biosynthetic pathway in cancer cells have not been fully uncovered. However, a recent and extensive analysis for thousands of tumors and their corresponding normal tissues of origin has revealed that the choice of the NAD-producing pathway in cancer cells is based upon the healthy tissues from which they originate [37]. Accordingly, tumors were classified into the following two broad categories: PH-dependent tumors and salvage-dependent tumors where NAPRT gene amplification and NAMPT enhancer remodeling, respectively, are the key hallmarks. Tissues with high basal NAPRT expression become dependent on NAPRT for survival upon malignant transformation. On the other hand, salvage-dependent tumors originate from tissues that lack NAPRT expression, and consequently, their NAD supply is primarily hinged upon NAMPT. This conclusion was further substantiated by the observation that neither the overexpression of NAPRT in salvage-dependent cancer cells nor the overexpression of NAMPT in PH-amplified cancer cells has reversed their predestined NAD metabolic pathway [37]. Notably, when NAMPT was depleted in salvage-dependent tumors, they were still able to maintain the NAD supply through the alternative salvage NR-NMRK1 pathway, and dual NMRK1 and NAMPT inhibition resulted in more effective NAD reduction and significant tumor suppression in vivo [37]. This finding provides insights into the role of NMRK1 in mediating resistance to NAMPT inhibition. In this section, we describe how the rate-limiting enzyme in each NAD synthetic pathway is regulated in cancer cells (Table 1).

### 3.1. NAMPT Regulation

Several transcriptional and post-transcriptional mechanisms tightly regulate NAMPT expression and activity in tumors. Chowdhry and colleagues recently reported on a putative *NAMPT* enhancer located 65 kb upstream of the *NAMPT* transcription start site, which controls NAMPT expression and activity [37]. Chromatin immunoprecipitation and further experiments revealed that this *NAMPT* enhancer is marked by H3K27 acetylation, is bound by the transcription factors c-MYC and MAX that regulate its activity, and that it is required solely by salvage-dependent tumors for their survival [37]. Consistent with a role for c-MYC in NAMPT expression, an earlier study described a c-MYC–NAMPT–SIRT1 positive feedback loop, in which c-MYC directly interacts with the NAMPT promoter and induces NAMPT expression, which in turn leads to SIRT1 activation through enhanced NAD provision [38]. SIRT1, in turn, stabilizes c-MYC and enhances its transcriptional activity, and promotes tumorigenesis through the attenuation of p53 activity and the inhibition of c-MYC-induced apoptosis [38]. This c-MYC–NAMPT–SIRT1 positive feedback loop was found to be activated in colorectal carcinoma and its interruption was proposed as a viable therapeutic intervention [59,60]. Additionally, the high-mobility group A (HMGA1) protein was reported to be another protein regulating NAMPT expression through a different enhancer element [39]. The HMGA1–NAMPT–NAD signaling axis was shown to drive the proinflammatory senescence-associated secretory phenotype (SASP) via NAD-mediated enhancement of nuclear-factor kappa B (NF-κB) activity, to promote an inflammatory environment and to drive tumor progression [39]. Indeed, numerous cancer types overexpress HMGA proteins and their overexpression is often associated with poor prognosis [61]. On the contrary, the transcription factor and tumor suppressor forkhead box O1 (Foxo1) binds to the 5′-flanking region of the *NAMPT* gene and downregulates NAMPT expression in breast cancer cells, an effect that is reversed by the insulin–PI3K–AKT signaling pathway [42]. Additionally, NAMPT-AS “RP11-22N19.2”, a new promoter-associated long non-coding RNA (Lnc-RNA), epigenetically regulates NAMPT expression in triple-negative breast cancer (TNBC) [43]. NAMPT-AS activates NAMPT expression at the transcriptional and post-transcriptional level, and promotes tumor progression and invasiveness in TNBC [43]. Similarly, in gliomas, gastric cancer-associated transcript 3 (GACAT3), another long non-coding RNA, regulates NAMPT expression and promotes glioma progression by acting as a molecular sponge to miR-135a, inhibiting its interaction with its target, NAMPT [44]. Several studies report on NAMPT expression being regulated at the post-transcriptional level by microRNAs. Specifically, *NAMPT* mRNA was found to be a target of miR-381 [45], miR-206 [46], miR-494 [47], and miR-154 [48] in breast cancer cells, of miR-23b in melanoma [51], of mir-206 in pancreatic cancer [50] and miR-26b [49] in colorectal cancer. Generally, increased expression of these microRNAs was shown to suppress NAMPT expression and it was associated with reduced cancer cell viability, suggesting the potential use of these microRNAs as anti-cancer agents. In this context, we have recently shown that, in addition to being regulated at the gene level, NAMPT enzymatic activity can also be regulated by other enzymes. Specifically, we found that SIRT6 enhances NAMPT enzymatic activity through direct protein deacetylation, protecting cancer cells against oxidative stress [40]. Similarly, a previous study found that also SIRT1 deacetylates NAMPT, predisposing it to secretion in adipocytes [41]. Mesenchymal glioblastoma stem cells were found to preferentially upregulate NAMPT and NMMT expression through the transcription factor C/EBPβ, which interacts with *NAMPT* and *NNMT* gene regulatory regions. Of note, in these cells subtypes, NNMT induced a state of DNA hypomethylation and downregulated the expression of DNA methyltransferases in a methionine-dependent fashion [36]. Whether NNMT epigenetically affects NAMPT expression requires further studies.

### 3.2. NAPRT Regulation

An increased *NAPRT* gene copy number is the major hallmark in the PH-dependent tumors and their matched tissues of origin [37,62]. By contrast, tumors exist that show *NAPRT* promoter hypermethylation and thus, lose NAPRT expression [52]. The treatment of these tumors with NAMPT inhibitors (NAMPTi) results in synthetic lethality, which could not be reversed by adding NA due to the dysfunctional NA–NAPRT route [52]. In addition to promoter hypermethylation, other mechanisms that regulate *NAPRT* gene expression include alternative splicing and mutations in the transcription factor binding sites [63]. Mounting evidence indicates that NAPRT is a central regulator of NAD metabolism and a critical determinant of the therapeutic success of NAMPT inhibitors. For instance, we demonstrated that NAPRT-expressing ovarian and pancreatic cancers are resistant to NAMPT inhibition, but downregulation of NAPRT significantly depletes intracellular NAD stores and sensitizes these tumors to NAMPT inhibitors [62]. On the other side, cancers displaying genetic alterations that suppress NAPRT activity are exquisitely vulnerable to NAMPT inhibitors’ monotherapy. For example, carcinomas with isocitrate dehydrogenase 1 *(IDH1)* mutations, such as gliomas and sarcomas, tend to downregulate NAPRT levels through hypermethylation of the *NAPRT* promoter, thereby blocking the NA–NAPRT pathway, which makes them dependent on NAMPT for NAD replenishment [53]. As a direct result, NAMPT inhibition was shown to result in a metabolic crisis in these types of tumors, blunting NAD pools and causing AMPK-mediated autophagy and cytotoxicity [53]. In keeping with this notion, mutations in the protein phosphatase Mg2+/Mn2+-dependent 1D *(PPM1D)* gene in pediatric gliomas also drive *NAPRT* gene silencing through the hypermethylation of CpG islands in the *NAPRT* promoter, thus, again conferring unique sensitivity to NAMPT inhibitors [54]. Additionally, extreme susceptibility to NAMPT inhibition was also seen with gastric cancer cell lines, which show markers of epithelial-to-mesenchymal transition (EMT), where this EMT subtype was associated with loss of NAPRT expression [64]. Collectively, these findings emphasize that NAPRT expression could be a useful biomarker of sensitivity to NAMPT inhibitors.

### 3.3. QAPRT Regulation

Given that many cancer cells lack the expression of the complete chain of enzymes of the de novo NAD biosynthetic pathway, they are unable to use tryptophan as a precursor for NAD production [65]. Nevertheless, elevated expression of QAPRT, the rate-limiting enzyme in the de novo NAD pathway, has been reportedly associated with resistance to NAMPT inhibitors, but also to chemotherapeutic agents in multiple types of cancer [66,67,68,69,70]. For instance, glioma cells were found to express QAPRT and utilize QA as a NAD precursor to protect themselves from oxidative stress and NAMPT inhibition, thereby drawing attention to the de novo NAD synthesis pathway as a potential therapeutic target [70]. Similar to NAMPT regulation, the RNA molecules miR-654-3p and the Down syndrome cell adhesion molecule antisense RNA 1 (DSCAM-AS1) were recently found to control the expression of QAPRT in ovarian and breast cancer cells, respectively [56,57]. Alternatively, the transcription factor Wilms’ tumor protein 1 (WT1) is a positive regulator for QAPRT transcription in leukemia cells through binding to conserved regions in the *QAPRT* promoter [58].

## 4. Chemical Inhibitors of NAD Biosynthesis

As mentioned earlier, NAMPT is the bottleneck enzyme in the NAD salvage pathway. Elevated NAMPT expression levels were extensively reported in numerous solid and hematological malignancies, including pancreatic cancer [50], gastric cancer [71], prostate cancer [72], breast cancer [73,74], colorectal cancer [75,76,77], gliomas [78], ovarian cancer [79], melanoma [80], thyroid carcinoma [81], sarcomas [82], and lymphomas [83]. Indeed, NAMPT is implicated in driving pro-oncogenic and more aggressive phenotypes, and its overexpression has been associated with poor prognosis in different types of cancer [74,75,76,78,84,85]. In addition to existing as an intracellular protein (intracellular NAMPT, iNAMPT), NAMPT also gets secreted extracellularly and this form of the protein is known as extracellular NAMPT (eNAMPT), but also as visfatin or as pre-B-cell colony-enhancing factor (PBEF). In fact, eNAMPT was originally identified as PBEF, a modulatory cytokine during B-cell development [86]. The eNAMPT also exerts pro-oncogenic effects by modulating the tumor microenvironment, enhancing tumor metabolism, and promoting EMT [87,88]. In melanoma, NAMPT is secreted by the cancer cells and NAMPT silencing was shown to reduce tumor progression and was accompanied by lower levels of circulating eNAMPT [89]. In line with this finding, eNAMPT levels were increased in the mice bearing melanoma cells that acquired resistance to BRAF inhibitors as well as in patients with BRAF-mutated metastatic melanoma, and high eNAMPT levels showed a negative correlation with patients’ overall survival [90]. Patients with invasive prostate cancer also showed higher plasma eNAMPT levels and neutralizing circulating eNAMPT was proven to markedly attenuate cancer invasiveness in prostate cancer mice models [91].

Moreover, a proinflammatory function of eNAMPT has already been established [92]. Consistently, serum levels of eNAMPT were found to be elevated in inflammatory bowel disease (IBD) patients who showed resistance to anti-TNFα therapy, and eNAMPT neutralization, with an anti-eNAMPT monoclonal antibody, ameliorated acute and chronic colitis in experimental mice models [93]. Notably, NAPRT was also found to exist as an extracellular protein (eNAPRT), which mediates inflammation by binding to toll-like receptor 4 (TLR4) and by activating the NF-κB pathway [94]. Consistent with these findings, NAPRT serum levels were strikingly elevated in septic patients [94]. Whether, similar to eNAMPT, eNAPRT also plays a pro-oncogenic role remains to be understood. Nonetheless, primary clues from 312 cancer patients diagnosed with solid or hematological malignancies just indicated mild elevations in their serum eNAPRT levels as compared to healthy donors (median serum eNAPRT was 1.4  ±  0.07 ng/mL in healthy donors, 2.4  ±  0.15 ng/mL in cancer patients and 27.1  ±  4.9 ng/mL in septic individuals) [94]. For a broader overview of the roles of eNAMPT and eNAPRT, we refer the reader to other recent articles [87,95,96].

Given its pleiotropic role in cancer pathogenesis, NAMPT has long been considered an attractive therapeutic target for cancer treatment. In this section, we provide a brief overview of some of the key NAMPT inhibitors that have been discovered so far (Figure 2), as well as of other inhibitors of NAD biosynthesis reported over the last years. The advances in medicinal chemistry and the pharmacology of small molecule NAMPT inhibitors are beyond the scope of this review. They have recently been reviewed by Sampath et al. [97] and by Galli et al. [98].

### 4.1. Specific NAMPT Inhibitors

#### 4.1.1. FK866 (also known as APO866, (E)-Daporinad, and WK175)

It is the first chemical compound reported as an NAMPT inhibitor. It shows a potency in the low nanomolar range (cellular IC_50_ of about 1 nM) [99]. An early study from 2002 showed that FK866 markedly reduced intracellular NAD levels which, in turn, triggered delayed apoptotic cell death in human leukemia cells [100]. One year later, FK866 was demonstrated to cause NAD depletion and to induce apoptosis in HEPG2 liver cancer cells via NAMPT inhibition [99]. Based on kinetic studies, FK866 was initially identified to be a non-competitive NAMPT inhibitor [99]. Later on, crystallographic studies of the NAMPT–FK866 complex revealed how FK866 binds within the enzyme catalytic domain [101,102,103] and suggested that FK866 could be a tight-binding competitive NAMPT inhibitor [103]. Up until now, FK866 has been extensively employed in preclinical cancer research and demonstrated robust antineoplastic efficacy across a wide variety of solid and hematological cancers both in vitro and in vivo.

#### 4.1.2. CHS-828 (GMX1778)

It is a pyridyl cyanoguanidine anticancer agent that was developed by Leo Pharma AS. It was first reported in 1997 and 1999 where it demonstrated potent antitumor activity in breast cancer and lung cancer cell lines, and in mice experiments the treatment with CHS-828 resulted in regression of human breast cancer and lung cancer tumors [104,105]. In fact, CHS-828 was evaluated in clinical trials even before FK866 [106]. However, it was not until 2008 that Olesen et al. showed that CHS-828 kills cancer cells mainly through NAD depletion in a similar fashion to FK866 and suggested that CHS-828 acts as an NAMPT inhibitor [107].

#### 4.1.3. GMX1777 (EB1627)

It is a prodrug of the compound CHS-828 that was developed in 2005 to improve the pharmacokinetic and solubility profile of the parent compound [108]. GMX1777 showed potent in vivo tumor-killing action as a single agent and potentiated the effect of etoposide in small-cell lung cancer mice models [108].

#### 4.1.4. OT-82

It is a novel NAMPT inhibitor developed by OncoTaris, Inc. It is a lead compound identified through the cell-based high-throughput screening of chemical libraries comprising more than 200,000 small molecules, followed by hit validation and structural optimization [109]. Its activity was further assessed in a cell panel of 12 hematological and 17 non-hematological malignancies. OT-82 demonstrated stronger activity towards hematopoietic malignancies with an average IC_50_ of 2.89  ±  0.47 nM compared to an average IC_50_ of 13.03  ±  2.94 nM in non-hematopoietic cancers [109]. OT-82 was shown to cause NAMPT inhibition with subsequent NAD and ATP depletion, and to induce apoptotic cell death [109]. For optimal OT-82 efficacy, the authors recommended adherence to diets that do not contain more than 100% of the recommended daily amount of niacin [109].

Besides, additional examples of reported NAMPT inhibitors include GNE-617 [110] and GNE-618 (Genentech) [111], MV87 [112], A-1293201 and A-1326133 (AbbVie) [113], STF-118804 [114], LSN3154567 (Eli Lilly) [115], and antibody–drug conjugates (ADCs) with NAMPT inhibitors [116,117].

### 4.2. Dual NAMPT Inhibitors

#### 4.2.1. KPT-9274 (ATG-019)

It is a dual inhibitor of NAMPT and of the serine/threonine p21-activated kinase 4 (PAK4) that was developed by Karyopharm Therapeutics. Its anticancer activity was primarily tested in human renal cell carcinoma where KPT-974 reduced cancer cell viability, invasion and migration, and induced apoptosis [118]. The downstream effects of PAK4 inhibition included reduced G2-M transition, nuclear β-catenin, and downregulation of c-MYC and cyclin D1. NAMPT inhibition caused significant NAD depletion and downregulation of SIRT1 activity [118]. KPT-9274 showed minimal cytotoxicity in vitro on normal primary renal cells and no significant weight loss in mice [118]. It is currently evaluated in phase I trials enrolling patients with solid tumors or lymphomas.

#### 4.2.2. STF-31

It is a hybrid inhibitor of NAMPT and glucose transporter 1 (GLUT1). It was initially reported to specifically bind to GLUT1, which in turn impairs glucose uptake and confers synthetic lethality against renal cell carcinomas that show loss of the von Hippel–Lindau (VHL) tumor suppressor gene, and become highly reliant on elevated glucose uptake and aerobic glycolysis [119]. Later on, cancer cell line profiling and genomic profiling of compound-resistant clones identified NAMPT as the target enzyme of STF-31 [120].

#### 4.2.3. Chidamide

It is a histone deacetylase (HDAC) inhibitor that is used for the treatment of cutaneous T-cell lymphoma [121]. Interestingly, NAMPT was very recently found to be a novel target of chidamide where it demonstrated NAMPT inhibitory activity (IC_50_ = 2.1 µM) and reduction in NAD levels in HCT116 cells [122]. This finding introduces a new mode of action of chidamide as a dual NAMPT/HDAC inhibitor. Notably, a number of dual NAMPT/HDAC inhibitors have been developed in earlier studies using pharmacophore fusion approaches [123,124]. Particularly, one compound (7f) demonstrated potent dual-target inhibition in the low nanomolar range [124].

### 4.3. Inhibitors of Other NAD-Producing Enzymes

#### 4.3.1. Vacor

It is an old rat poison [125]. It was recently shown to have cytotoxic activity against NMNAT2-expressing cancer cells [126]. Mechanistically, Vacor is converted via NAMPT and then via NMNAT2 into the NAD analog, Vacor adenine dinucleotide (VAD) and Vacor metabolism results in the inhibition of NMNAT2, NAMPT, and also NAD-dependent dehydrogenases [126]. This extensively and immediately depletes NAD, impairs glycolysis, prompts energy failure, and ultimately kills NMNAT2-proficient cancer cells by necrosis [126]. In vivo, Vacor suppressed tumor growth of NMNAT2-expressing xenografts of neuroblastoma and melanoma [126]. Similarly, gallotannin was also found to be a potent inhibitor of all NMNAT isoforms, with NMNAT3 being the most sensitive (IC_50_ = 2 µM) followed by NMNAT1 (IC_50_ = 10 µM) and finally NMNAT2 (IC_50_ = 55 µM) [16].

#### 4.3.2. 2-Hydroxy Nicotinic Acid (2-HNA)

It is a competitive inhibitor of NAPRT with an apparent inhibitory constant (Ki) of 0.23 mM [127]. It was primarily identified as a potent structural analog of nicotinic acid, which inhibited the accumulation of radioactivity in blood platelets incubated with radiolabeled [_14_C^7^] nicotinic acid (10 µM) [128]. It showed 95% inhibition at 1 mM with a direct relationship between the extent of inhibition and the 2-HNA concentration [128]. In NAPRT-expressing ovarian and pancreatic cancer cells, 2-HNA recapitulated the effects of NAPRT silencing and sensitized cells to FK866 [62]. In addition to 2-HNA, other reported NAPRT inhibitors include non-steroidal anti-inflammatory compounds such as flufenamic acid, salicylic acid, mefenamic acid, phenylbutazone, and indomethacin [129]. Furthermore, NAPRT was reported to be inhibited by several metabolites involved in glucose and fatty acid metabolisms such as glyceraldehyde 3-phosphate, phosphoenolpyruvate, fructose 1,6-bisphosphate, Co-A, acetyl-CoA, glutaryl-CoA, and succinyl-CoA. Of note, Co-A was the most effective among them and manifested an IC_50_ of about 0.85 mM [130].

#### 4.3.3. N-(3,4-dichlorophenyl)-4-{[(4-nitrophenyl)carbamoyl]amino}benzenesulfonamide (Compound 5824)

It is an inhibitor of the bacterial NAD synthetase, the enzyme that catalyzes the amidation of NAAD to NAD in the PH pathway. Brouillette’s group primarily focused on developing chemical inhibitors of the bacterial NAD synthetase (NadE) for their potential use as antibacterial compounds [131,132]. In silico screening and docking studies followed by high-throughput assays of enzyme inhibition and antibacterial activity identified the molecule 5824 as a lead compound acting as a bacterial NAD synthetase inhibitor (IC_50_ = 10 µM) [133]. Given that compound 5824 was suggested to bind to the NAAD binding site in the NAD synthetase of *B. subtilis* [133,134], which was found to share a very high degree of residue conservation with the NAAD binding site of the human NAD synthetase 1 (NADSYN1), Chowdhry and colleagues employed compound 5824 as a human NADSYN1 inhibitor in cancer models [37]. Interestingly, this compound showed a significant and dose-dependent reduction in NAD levels and suppressed the growth of PH-amplified (OV4) xenografts but not of salvage-dependent (H460) tumors [37].

## 5. Effects of NAD Production Inhibition in Cancer

Deregulating NAD homeostasis in the cancer cells through NAD synthesis inhibitors was shown to result in antiproliferative and cytotoxic effects via different mechanisms. In this section, we report the downstream effects of NAD deprivation in cancer cells and combination strategies that were shown to potentiate the NAMPT inhibitors, eliciting a more pronounced antitumor response (Table 2).

### 5.1. NAD Depletion and Cancer Cell Death

Several mechanisms were reported to trigger cancer cell death in response to NAD depletion by NAMPT inhibitors. Initially, FK866 was reported to deplete intracellular NAD and to kill cancer cells by inducing apoptosis [99,100]. Similarly, OT-82 and KPT-9274 were also found to induce their antileukemic effect through apoptosis [109,169]. In primary chronic lymphocytic leukemia (CLL) cells, FK866 was demonstrated to induce apoptotic signaling and caspase activation at doses that triggered cell death [135]. However, in the same models, indicators for autophagy induction were also observed at low doses of FK866 and/or at early time points [135]. Several studies linked NAMPTi-induced cell death to autophagy. In multiple myeloma cells, FK866 triggered autophagic cell death via (i) the inhibition of PI3K/mTORC1 activity (a transcription-independent mechanism) and (ii) the inhibition of MAP kinase (MAPK), which permits nuclear translocation of the transcription factor EB (TFEB) that coordinates lysosomal biogenesis and drives the expression of autophagy-related genes [170,171]. Additionally, no evidence of apoptotic cell death was detected in multiple myeloma cells in response to FK866 treatment [170]. Consistent with this observation, autophagy, but not apoptosis, was associated with FK866-induced cytotoxicity in neuroblastoma and hematological cancers [172,173]. In neuroblastoma cells, FK866-induced autophagy was potentiated or antagonized by chloroquine and by 3-methyl adenine (3-MA), respectively, lending support to the notion that aberrant autophagy is involved in FK866-mediated cancer cell demise [138]. Furthermore, NAMPTi-induced NAD depletion in *IDH1*-mutant cancers was associated with inducing AMP kinase (AMPK) and initiating autophagy, and the autophagy inhibitor 3-MA rescued the cells from the cytotoxic effects of NAMPTi-mediated NAD depletion [53]. Intriguingly, FK866 simultaneously activated apoptosis and autophagy and markedly reduced the viability of HTLV-1-infected, adult T-cell leukemia/lymphoma (ATL) cell lines [174]. Furthermore, we showed that autophagy-mediated FK866 antileukemic activity was potentiated by tumor necrosis factor-related apoptosis-inducing ligand (TRAIL) [162]. Another group postulated that oncosis is the critical pathway leading to cancer cell death in response to the NAMPT inhibitor, GNE-617, in several non-hematological cancer cell lines irrespective of the appearance of signs of apoptosis or autophagy [175]. This group also showed that oncosis was driven by dramatic ATP depletion and subsequent loss of plasma membrane integrity, which typically marks the late phases of NAD depletion via NAMPT inhibition [175]. Taken together, the mechanism underlying cancer cell death in response to NAMPT inhibitors might be cancer type-specific and regulated in a dose and time-dependent manner.

### 5.2. NAD Depletion and Oxidative Stress

Oxidative stress is caused by an imbalance between reactive oxygen species (ROS) and the cellular antioxidant capacity in favor of the former. Excessive ROS accumulation is detrimental to cell viability. NADPH is a critical molecule in oxidative homeostasis as it provides the reductive power for glutathione reductase and thioredoxin reductase in glutathione and thioredoxin ROS scavenging systems [10,176]. NADPH is mainly produced via the PPP and accordingly, cancer cells have evolved mechanisms to enhance glucose flux into the PPP to combat oxidative stress [177,178]. As a consequence, building blocks for nucleotide biosynthesis are also more available to cancer cells. Besides, around 10% of the total NAD(H) pool is phosphorylated to NADP(H) by NAD kinases [10,27]. Several studies described a strong link between NAD inhibition and oxidative stress in cancer cells. For instance, enhanced ROS production in MDA-MB-231 breast cancer cells was noted when FK866 was added to ROS-containing plasma-activated medium (PAM) [179]. Also, recently FK866 was reported to exert its antileukemia activity through ROS and reactive nitrogen species generation as a consequence of NAD depletion [136]. Mitochondrial depolarization, ATP loss, and cell death were also reported as downstream effects of FK866-induced oxidative stress in this study [136]. Increased ROS levels were also reported with other NAMPT inhibitors, such as GMX1778 and OT-82 [154,180]. In support of the above insights, combining FK866 with β-lapachone, an NQO1 substrate that generates ROS and exerts anticancer effects, was shown to cause dramatic NAD depletion and cytotoxic effects in NQO1-expressing pancreatic ductal adenocarcinoma (PDAC) and non-small cell lung cancer (NSCLC) cells [164,165,166]. Curiously, in renal and in cochlear tissues, β-lapachone was reported to augment NAD levels and to reverse the drop in the NAD/NADH ratio caused by cisplatin treatment. [28,181]. This different and apparently contrasting behavior of β-lapachone seems to be attributable to the preferential accumulation of ROS species via this compound in NQO1-overexpressing cancers, but not in normal cells (that are protected by low NQO1 expression and by high catalase levels). In NQO1-overexpressing cancer cells, ROS production in response to β-lapachone causes DNA damage and thereby triggers PARP-mediated NAD degradation to such an extent that it outweighs the possible increase in NAD caused by NQO1, ultimately causing cancer cell demise via NAD and ATP shortage [182,183]. Similarly, paracetamol was found to bind to NQO2 as an off-target effect and NQO2 expression modulated paracetamol-induced ROS production in HeLA cells [184]. Therefore, combining paracetamol with NAD-depleting agents in NQO2-overexpressing cancers might be a promising approach. Last but not least, NAMPT inhibitors augmented oxidative stress induced by temozolomide (TMZ) in glioblastoma cells, and this sensitization effect was reversed by the ROS scavenger tocopherol [149].

### 5.3. NAD Depletion and DNA Damage and Repair

The crosstalk between NAMPT, PARPs, and DNA damage has been thoroughly investigated over the past decades. While NAMPT produces NAD, activated PARPs consume the majority of NAD to support their DNA repair activity in response to DNA-damaging insults [10]. In line with this notion, excision repair cross-complementation group 1 (ERCC1)-deficient NSCLC cells, a DNA-repair defective cancer model, showed reduced basal NAMPT and NAD levels, presumably as a result of chronic PARP1 activation, and FK866 treatment resulted in a catastrophic NAD drop and profound synthetic lethality in vitro and in vivo [185]. Chemo-potentiation was seen when NAMPT inhibitors were combined with agents that cause DNA damage, such as 5-fluorouracil (5-FU) in gastric cancer [71], fludarabine and cytarabine in leukemia [135,154], cisplatin or etoposide in neuroblastoma [138], temozolomide in gliomas [149,150], gemcitabine, paclitaxel and etoposide in pancreatic cancer cells [144,145], pemetrexed in NSCLC [143], and bendamustine and melphalan in Waldenstrom macroglobulinemia (WM) [159]. In addition, NAMPT inhibitors potentiated the efficacy of the radionuclide ^177^Lu-DOTATATE in neuroendocrine tumors [147], and sensitized head and neck cancer and prostate cancer models to radiotherapy [186,187]. Also, it was hypothesized that combining NAMPT inhibitors with PARP inhibitors would further downregulate PARP activity leading to persistent DNA lesions and ultimately cell death. Indeed, a potentiating effect was shown between NAMPT inhibitors and the PARP inhibitors olaparib or niraparib in triple-negative breast cancer and Ewing sarcoma [151,152,153]. In opposition, a recent study demonstrated that the FK866 cytotoxic effect against hematological malignant cells is reliant on PARP integrity since PARP1 deletion reversed ROS accumulation, mitochondria depolarization and ATP loss, and abolished FK866-induced cell death [136]. Similar results were previously reported by our group in human activated T-cells and T-cell acute lymphoblastic leukemia (ALL) models [188]. The proposed explanation for these findings is that PARP inhibitors, by blocking NAD consumption, elevate NAD levels and this antagonizes NAMPTi-induced NAD depletion and its downstream effects.

### 5.4. NAD Depletion and Targeted Therapy

Owing to their particular mode of action, NAMPT inhibitors lend themselves to be used in combination regimes, enhancing the antitumor activity of targeted therapies such as histone deacetylase inhibitors [146] and tyrosine kinase inhibitors in leukemia [154] and WM [142], proteasome inhibitors in multiple myeloma [141], and mTOR inhibitors in pancreatic neuroendocrine tumors [160]. Finally, we demonstrated that cyclosporin-A and verapamil sensitized leukemia cells to FK866 by inhibiting P-glycoprotein 1 (Pgp), the multidrug resistance transporter, thereby permitting the intracellular accumulation of FK866, which in turn led to ER stress and cell demise [140].

## 6. In Vivo Studies of NAD Production Inhibitors in Mice

### 6.1. Efficacy of NAMPT Inhibitors In Vivo

The short-term treatment with NAMPT inhibitors, including multitarget NAMPT inhibitors or NAMPTi-ADCs, displayed marked antineoplastic efficacy across a wide spectrum of tumor murine models, either as single-agent therapy or in combination with other anticancer treatments.

For instance, FK866 demonstrated antitumor, antiangiogenic, and antimetastatic effects in renal cancer murine models [189]. GNE-617 showed robust antitumor activity in NAPRT-deficient xenograft mouse models, including prostate (PC3) cancer, fibrosarcoma (HT1080), and pancreatic (MiaPaCa-2) cancer, and resulted in more than 98% of the NAD reduction in tumors in vivo [190]. Likewise, its structurally related derivative, GNE-618, demonstrated to suppress tumor growth in A549 NSCLC xenografts as well as in patient-derived gastric cancer and sarcoma xenografts [65,190].

Given that *IDH1*-mutant cancers are exquisitely reliant on NAMPT for their NAD supplies (as they downregulate NAPRT expression), NAMPT inhibitors were found to exhibit remarkable antitumor activity against *IDH1*-mutant glioma and fibrosarcoma xenografts [53]. Alternatively, a recent study aimed to achieve NAD depletion in *IDH*-mutant cancer cells, not by interfering with NAD synthesis but by blocking the regeneration of the mono ADP-ribose moieties from poly(ADP-ribose) (PAR) through the inhibition of poly(ADP-ribose) glycohydrolase (PARG), the enzyme responsible for PAR breakdown, with the simultaneous enhancement of PARP-mediated NAD consumption via DNA-damaging agents administration. As hypothesized, in an *IDH*-mutant fibrosarcoma (HT1080) mouse model, temozolomide combination with PARG knockout depleted freely available NAD by preventing PAR breakdown, leading to hyperPARylation, NAD sequestration, metabolic collapse and, indeed, significant tumor growth suppression was noted [191].

Although cellular senescence is an antitumor mechanism, the so-called senescence-associated secretory phenotype (SASP) sees senescent cells secrete several pro-inflammatory and pro-angiogenic factors, and was implicated in creating a tumor-promoting microenvironment and promoting cancer stemness [192,193]. In mice bearing epithelial ovarian cancer xenografts, the addition of FK866 to cisplatin prolonged mouse survival and delayed the outgrowth of cisplatin-treated tumors upon treatment cessation [139]. NAMPT inhibition was shown to mediate this additional anticancer benefit by i) NAD-mediated inhibition of the cisplatin-induced SASP in epithelial ovarian cancer and ii) suppression of the platinum-induced senescence-associated cancer stem cells [139].

Concerning the antileukemic activity of NAMPT inhibitors, OT-82 showed in vivo efficacy against hematological malignancies in a dose-dependent manner [109]. OT-82 suppressed the tumor growth of subcutaneous xenografts of acute myeloid leukemia (AML) (MV4-11), erythroleukemia (HEL92.1.7), Burkitt lymphoma (Ramos), and multiple myeloma (RPMI 8226), and prolonged survival of mice with systemic xenografts of AML (MV4-11), erythroleukemia (HEL92.1.7), infant MLL-arranged ALL (MLL-2) and with patient-derived xenografts (PDX) of high risk ALL [109,154]. In the latter model, OT-82 was found to delay leukemia growth in 95% (20/21), and cause disease regression in 86% (18/21), of the pediatric ALL PDXs [154]. Additionally, OT-82 alone showed comparable efficacy to an induction-type chemotherapeutic regimen used to treat pediatric ALL and improved the efficacy of cytarabine and dasatinib against pediatric ALL in mice [154]. Consistent with these reports, earlier studies had shown that FK866 elicited potent in vivo antitumor activity in human xenograft models of ATL, AML, Burkitt leukemia, and lymphoma [173,174]. Additionally, in a Burkitt lymphoma (Ramos) xenograft model, combining FK866 with rituximab resulted in prolonged mouse survival and reduced tumor burden as compared to mice receiving the single agents [148]. In addition, FK866 plus bortezomib were shown to act synergistically in multiple myeloma (MM1S) xenograft models [141].

OT-82 was also reported to reduce tumor growth and to prolong mouse survival in Ewing sarcoma xenograft (TC71 and TC32)-bearing mice, although tumors were found to grow again upon treatment discontinuation [153]. Notably, the combination of low doses of OT-82 and drugs that augment DNA damage, such as irinotecan or niraparib, improved the anticancer efficacy of OT-82 in orthotopic xenografts (TC32) and patient-derived xenografts of Ewing sarcoma [153].

The novel NAMPT/PAK4 inhibitor, KPT-9274, elicited remarkable antitumor effects in a broad panel of tumor mouse models, including models of renal cell carcinoma [118], B-cell acute lymphoblastic leukemia [194], acute myeloid leukemia [169], melanoma [158], colon cancer [158], triple-negative breast cancer [195], pancreatic ductal adenocarcinoma [161], pancreatic neuroendocrine tumors [160], rhabdomyosarcoma [196], Ewing sarcoma [197], multiple myeloma [198], and Waldenstrom macroglobulinemia [159]. It should be mentioned that the antitumor activity of KPT-9274 in rhabdomyosarcoma, Ewing sarcoma, multiple myeloma, breast cancer, pancreatic ductal adenocarcinoma, colon carcinoma, and melanoma was shown to be substantially driven by PAK4 inhibition [158,161,195,196,197,198,199]. Overall, compelling results from preclinical studies of KPT-9274 and OT-82 (summarized in Table 3) built a strong rationale for the evaluation of these two inhibitors in the clinic.

Combining NAMPT inhibition with cancer immunotherapy (e.g., by anti-PD1 antibodies) has been put forward as a promising therapeutic opportunity. This, in the first place, in view of studies showing that once secreted in the extracellular environment, NAMPT promotes the polarization of macrophages towards the “immunosuppressive” M2 phenotype [200]. In glioblastoma-bearing mice, the local delivery of microparticles loaded with the NAMPT inhibitor GMX1778, combined with a systemic anti-PD1 checkpoint blockade, resulted in higher T-cell recruitment and prolonged mice survival compared to monotherapy [155]. Of note, one animal survived 100 days and was confirmed tumor-free at the end of the experiment. The mice bearing renal carcinoma and melanoma tumors demonstrated an improved antitumor response when cotreated with anti-PD1 and KPT-9274 in comparison with either anti-PD1 or KPT-9274 monotherapy [157,158]. Interestingly, in the renal carcinoma model, KPT-9274 monotherapy was found to significantly reduce NAD + NADH levels, while the combined KPT-9274 and anti-PD1 treatment did not result in an overall reduction in the total NAD + NADH levels [157]. In support of the above findings, enhanced antitumor activity was seen when combining the NAMPT inhibitor MV87 and an anti-PD1 antibody compared to the single therapies in a fibrosarcoma model [156]. Strikingly, NAD replenishment through exogenous NMN supplementation sensitized anti-PD-(L)1 therapy-resistant (PD-L1 low) tumors to immunotherapy in mice models representing immunotherapy-tolerant lung cancer, pancreatic cancer, and hepatic cancer [201]. The latter effect was ascribed to the finding that NAD metabolism drives interferon γ (IFNγ)-induced PD-L1 expression in tumor cells via an activated NAMPT–TET1–pSTAT1–IRF1–PD-L1 axis, which in turn promotes tumor immune invasion in a CD8+ T cell-dependent manner [201]. However, this finding is in contradiction with another study that reported that NAD depletion via NAMPT inhibitors upregulated PD-L1 expression in glioblastoma cells [155]. Altogether, these results imply that further understanding of the crosstalk between NAMPT and PD-L1 is warranted.

With respect to the development of multitarget NAMPT inhibitors, compounds 35 and 7f are two recently discovered dual NAMPT/HDAC inhibitors, which demonstrated superior or comparable antitumor efficacy in comparison with FK866 or vorinostat monotherapy in HCT116 xenograft mice models [123,124]. Very recently, Zhang et al. applied molecular hybridization techniques using CHS-828 and erlotinib as templates and designed several first-in-class compounds that inhibit both NAMPT and epidermal growth factor receptor (EGFR) simultaneously [202]. Notably, the most active NAMPT/EGFR inhibitor showed a potent reduction in cell viability across several cancer cell lines and markedly suppressed tumor growth in a human H1975 lung cancer mouse model [202].

Lastly, the single-dose administration (20 mg/kg) of two novel anti-c-KIT antibody–drug conjugates, with novel NAMPT inhibitors as payloads, efficiently blocked in vivo tumor proliferation in c-Kit-positive gastrointestinal stromal tumor GIST-T1 mouse xenografts [117]. In agreement with these results, another group reported the strong in vivo antitumor activity of NAMPTi-ADCs in xenograft models of AML, Hodgkin lymphoma, and non-Hodgkin lymphoma [116].

### 6.2. Impact of NA on the Efficacy of NAMPT Inhibitors In Vivo

While normal tissues express NAPRT and can utilize NA to produce NAD, many cancer cells lack the PH pathway for NAD biosynthesis. Consequently, coadministration of NA with NAMPT inhibitors was proposed as a strategy to mitigate the off-target toxicity of NAMPT inhibitors, thereby widening their therapeutic index. According to this approach, NAPRT expression within tumor cells is the key determinant of the cancer types that might benefit from it [52]. Indeed, NA supplementation was shown to rescue NAPRT-proficient cancer cell lines from NAMPTi-induced cytotoxicity, while it was unable to reverse NAMPTi-induced toxicity in NAPRT-deficient cells [52,53,64,115,118,180,190,203].

Nevertheless, the results from mice experiments with NAPRT-negative tumors reflected a more complex situation. In agreement with the aforementioned in vitro results, early studies reported that NA cotreatment abolished the antitumor effects of FK866, GNE-617, and GMX1777 in mice bearing xenografts of NAPRT-proficient tumors (ML-2 and HCT-116) [190,203,204], while not affecting the efficacy of GMX1777 (24 h IV infusion 150 mg/kg and 650 mg/kg) or LSN3154567 (20 mg/kg; BID) in xenografts of NAPRT-deficient cancer cells (HT-1080 and NCI-H1155) [115,203]. However, in other experiments, the in vivo antitumor effects of the NAMPT inhibitors, GNE-617 and GNE-618, were found to be abrogated by NA supplementation even in the xenografts of NAPRT-deficient cell lines and patient-derived xenograft tumor models [190]. This rescue effect of supplemented NA was proposed to be achieved through partial NAD replenishment in the tumors, occurring independently of the PH pathway [190]. In another study, NA reversed the antiproliferative effect of FK866 in NAPRT-deficient A2780 xenografts at the standard 15 mg/kg dose, and the sensitivity to FK866 in the presence of NA was restored at the 50 mg/kg dose, which demonstrated an even more significant anticancer benefit compared to the standard FK866 (15 mg/kg) regimen [204]. Interestingly, in a multiple myeloma (IM-9) model, NA abrogated the antitumor effect of GMX1777 (150 mg/kg) when infused immediately after the end of the GMX1777 infusion, but when it was administered after 24 h from the end of GMX1777 infusion, the antitumor activity of GMX1777 persisted [205]. More recently, Shats and colleagues reported that mycoplasma-contaminated colon cancer cells were resistant to the NAMPT inhibitor STF-118804 in culture, but also in vivo once xenografted into nude mice, and a similar effect was seen with *E.coli* in vitro [25]. The authors proved that cancer cells were protected from NAMPTi-induced NAD depletion by the ability of the bacteria to convert NAM into NA, via their bacterial enzyme nicotinamidase (PncA), and thereby supply the precursors of the PH pathway [25]. Likewise, gut microbiota were demonstrated to deamidate exogenous NAM to NA and mediate NAD production in mammalian tissues, postulating that the microbial flora could mediate tumor resistance to NAMPT inhibitors through endogenous NA provision and PH pathway engagement [25]. In addition, it was found that the NA content of a diet is a key determinant of NAMPT inhibitor efficacy. For instance, one study showed that the efficacy of OT-82 was inversely correlated with the niacin (NA) dietary content in leukemia-bearing mice [109]. Thus, the complex systemic coordination of NAD production, as well as the metabolite exchange mechanisms, make so that the in vivo effects of NAD biosynthesis inhibition strategies can hardly be anticipated. In addition, albeit NA supplementation could potentially enhance the tolerability of NAMPT inhibitors, caution must be taken with this intervention as NA might, at least in part, attenuate their antitumor action.

### 6.3. Toxicity of NAMPT Inhibitors In Vivo

The adverse effects associated with NAMPT inhibitors have been comprehensively investigated in animal models. In an acute toxicological study, treatment of mice with high doses of FK866 (60 mg/kg bid for 4 days) resulted in thrombocytopenia and severe lymphopenia which were reverted by NA cotreatment [204]. In the same study, histological signs of drug-related toxicities were observed in the testis and spleen but not in the retina, lungs, heart, or brain. Similar hematological results were obtained in another study, which showed that rats treated with various NAMPT inhibitors, including FK866 and GM1778, for periods up to 15 days demonstrated lymphopenia and reticulocytopenia (reduced lymphocytes and reticulocytes counts) but not thrombocytopenia [206].

In non-tumor-bearing mice, NA infusion reduced the mortality associated with the administration of toxic doses of GMX1777 (650 mg/kg and 750 mg/kg) [203,205]. Similarly, NA protected B6D2F1 mice from toxic doses of FK866 [204]. These results strengthen the rationale for using NA as an antidote for NAMPT inhibitors and to widen their therapeutic index.

GNE-617 and GMX1778 were associated with retinal toxicity and fatal cardiac toxicity in short-term safety studies in rodents, and these effects were reproducible in human- and rat-relevant cell systems [97,207,208]. Of note, systemic NA treatment did not mitigate the retinal toxicity associated with GNE-617 and GMX1778 in rodents and only partially protected them from the NAMPTi (GNE-617)-induced cardiotoxicity [207,208]. The NAMPT inhibitors A-1326133 and A-1293201 were also reported to cause retinal damage in rats and dogs [209]. On the other hand, LSN3154567 did not cause retinopathy in rats treated with 20, 40, and 80 mg/kg of this compound for 4 days [115]. Yet, retinal toxicity was still observed in dogs that received LSN3154567 (1 and 2.5 mg/kg/day) for 4 days [115]. As opposed to what was observed with GMX1778 and GNE- 617, the co-administration of NA with LSN3154567 (5 mg/kg) did protect the dogs from retinopathy [115]. Altogether, these findings suggest that NAMPTi-induced retinal damage and its mitigation by NA could be chemotype-specific and might show species variability depending on the used animal model. Lastly, the recently reported NAMPT inhibitor OT-82 showed a favorable toxicological profile with no cardiac, neurological, or retinal toxicities in mice and cynomolgus monkeys (non-human primates), thus apparently avoiding the side effects of other NAMPT inhibitors [109]. Noteworthy, a combined low-dose OT-82 and niraparib therapy for one month led to several unexpected deaths in Ewing sarcoma-bearing mice, raising the concern of potential toxicities associated with prolonged combination treatments with this compound [153]. Finally, NAMPTi-ADCs were shown to have a remarkably improved toxicological profile. For example, IgG-8, a non-binding NAMPTi-ADC, was well tolerated at a dose of 100 mg/kg in a single-dose rat toxicity study and was only associated with mild acute cytopenias that were recovered four weeks post-injection with no reported thrombocytopenia, retinal toxicity, or cardiac toxicity [116]. This dose is 10-folds higher than the dose of the targeted ADCs (αCD19-8 and αCD30-8) at which antitumor activity was typically seen [116].

## 7. Perspectives and Obstacles for Clinical Uses of NAD Biosynthesis Inhibitors

Based on the encouraging preclinical data, the early-generation NAMPT inhibitors, FK866 (APO-866), GMX1778 (CHS-828), and its prodrug GMX1777, were investigated in early-phase clinical trials. Thrombocytopenia was the common dose-limiting toxicity associated with the three NAMPT inhibitors [210,211,212]. Additionally, several gastrointestinal (GI) adverse effects and high intra- and inter-patient pharmacokinetic variability were reported with CHS-828 [106,210,213]. GI hemorrhage and skin rash were other dose-limiting toxicities that accompanied GMX1777 [211,214]. Regrettably, the lack of objective tumor response and unfavorable side effect profile upon treatment with NAMPT inhibitors have halted their further progression in clinical trials so far [106,210,211,212,213,214,215].

The failure of NAMPT inhibitors in clinical trials has fueled research endeavors to overcome the limited clinical outcome of these agents. Indeed, numerous approaches have been sought to reduce their toxicities and re-sensitize cancer cells to NAMPT inhibitors, which can be summarized as the following (Figure 3):
(i)Developing safer and more effective “next-generation” NAMPT inhibitors;(ii)Using NAMPT inhibitors against a subset of cancers that show unique sensitivity to NAMPT inhibitors, such as hematological cancers or *IDH*-mutant cancers. NA might be administered simultaneously to protect normal tissues (circumventing NAMPT inhibition by the PH pathway to sustain adequate NAD stores). Although NA might alleviate the systemic toxicity and widen the therapeutic index of NAMPT inhibitors, abrogation of antitumor efficacy could represent a caveat to this approach;(iii)Combining NAMPT inhibitors with chemotherapy, immunotherapy, or radiation to achieve a synergistic effect;(iv)Combining NAMPT inhibitors with NAPRT inhibitors against NAPRT-positive cancer subtypes [62];(v)Development of “broad-spectrum” NAMPT inhibitors such as the dual NAMPT-PAK inhibitors, NAMPT-HDAC inhibitors, NAMPT-GLUT1 inhibitors, and recently the NAMPT-EGFR inhibitors;(vi)Development of NAMPTi-ADCs that selectively target NAMPT in cancer cells through antibody binding to cancer-specific cell surface markers, thereby sparing the normal cells from systemic NAD depletion.

Notably, two compounds of the second wave of NAMPT inhibitors (OT-82 and KPT-9274) are being currently investigated in phase I trials. In a two-stage (dose escalation and dose expansion) phase 1 study, the safety and efficacy of OT-82 are being evaluated in participants with relapsed or refractory lymphoma (NCT03921879). In the PANAMA (NCT02702492) and TEACH (NCT04281420) phase 1 trials, the safety, efficacy, and tolerability of KPT-9274 will be evaluated in patients with advanced solid tumors and malignancies or non-Hodgkin’s lymphoma. In both studies, the experimental arms include KPT-9274 alone or in combination with niacin ER. In the PANAMA trial, additionally, a third arm will also assess the combination of KPT-9274 and the immune checkpoint inhibitor nivolumab (anti-PD1). So far, no results have been disclosed.

## 8. Future Outlook

In summary, NAD-producing enzymes represent a collection of targets that could be exploited to combat cancer proliferation. Even though NAMPT is the most extensively NAD-producing enzyme targeted by chemical inhibitors, limited objective responses and unacceptable toxicities in the clinical trials have hampered faster progressions in the field. Given its pleiotropic roles in energy metabolism, promoting DNA repair in response to genomic insults, epithelial to mesenchymal transition, oncogenic signaling, and suppressing the immune surveillance within the tumor milieu, NAMPT remains an attractive target in cancer therapy [216]. As mentioned earlier, two NAMPT inhibitors are being currently evaluated in clinical trials. Nevertheless, in light of accumulating evidence that substantiated the significance of NAPRT expression as a parallel regulator of NAD metabolism in cancer and as a biomarker for NAMPT inhibitor therapy [37,62], the development of NAPRT inhibitors also represents a promising therapeutic approach. To our best knowledge, no potent NAPRT inhibitor has been identified so far. As a consequence, an important future challenge will be to develop NAPRT inhibitors with a higher potency than that of the currently available compounds (e.g., 2-hydroxy nicotinic acid) and with optimized drug-like properties. Additionally, given the recently emerging function of the gut microbiota as a key contributor to NAD metabolism, manipulation of the intestinal microbiome, presumably through antibiotics, might be exploited to modulate systemic and tumor NAD biosynthesis [25]. Furthermore, recent studies show that limiting dietary intake of niacin and tryptophan creates NAD deficiency in mouse models that recapitulate human physiology [31]. Consistent with this finding, curbing the content of NAD precursors in diets, together with NAD-depleting agents, might allow to more effectively lower NAD levels in cancer cells and thus, provide enhanced antitumor activity. In addition, processes such as polyunsaturated fatty acid (PUFA) desaturation and lactate fermentation prompt regeneration of NAD from NADH, and permit glycolytic NAD recycling and cell survival when the NAD/NADH ratio decreases in the cytosol [217]. These findings raise the possibility that such processes might be targeted to improve NAMPT inhibitors’ efficacy in malignant cells. Lastly, nicotinamide mononucleotide adenylyltransferase (NMNAT 1–3), nicotinamide riboside kinase (NMRK1/2), and NAD synthetase (NADSYN) stand out as additional understudied enzymes that warrant further exploration as targets in oncology.

## Figures and Tables

**Figure 1 nutrients-13-01665-f001:**
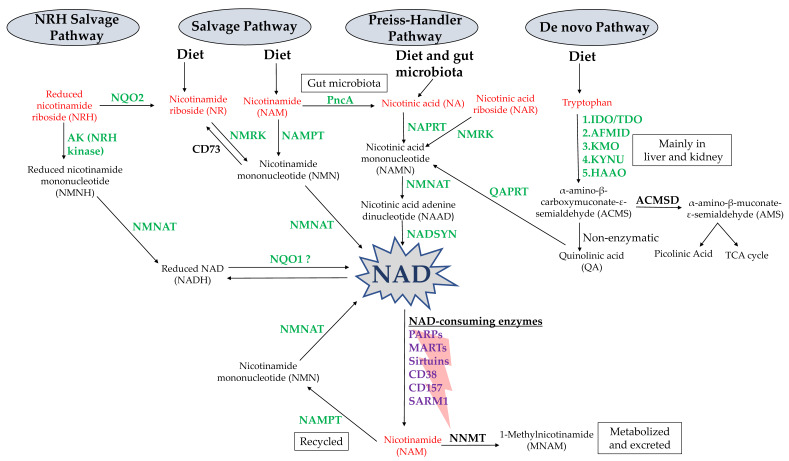
Schematic representation of the NAD biosynthetic pathways. NAD, nicotinamide adenine dinucleotide; NAMPT, nicotinamide phosphoribosyltransferase; NAPRT, nicotinic acid phosphoribosyltransferase; PncA, bacterial nicotinamidase; NMNAT, nicotinamide mononucleotide adenylyltransferase; NMRK, nicotinamide riboside kinase; NADSYN, NAD synthetase; QAPRT, quinolinic acid phosphoribosyltransferase; IDO, indoleamine-2,3-dioxygenase; TDO, tryptophan-2,3-dioxygenase; AFMID, arylformamidase; KMO, kynurenine 3-monooxygenase; KYNU, kynureninase; HAAO, 3-hydroxyanthranilate 3,4-dioxygenase; ACMSD, α-amino-β-carboxymuconate-ε-semialdehyde decarboxylase; MARTs, mono(ADP-ribosyl) transferases; PARPs, poly(ADP-ribose) polymerases; SARM 1, sterile alpha and TIR motif-containing 1; NQO1, NAD(P)H:quinone oxidoreductase; NQO2, NRH:quinone oxidoreductase; AK, adenosine kinase; NNMT, nicotinamide N-methyltransferase; and TCA, tricarboxylic acid.

**Figure 2 nutrients-13-01665-f002:**
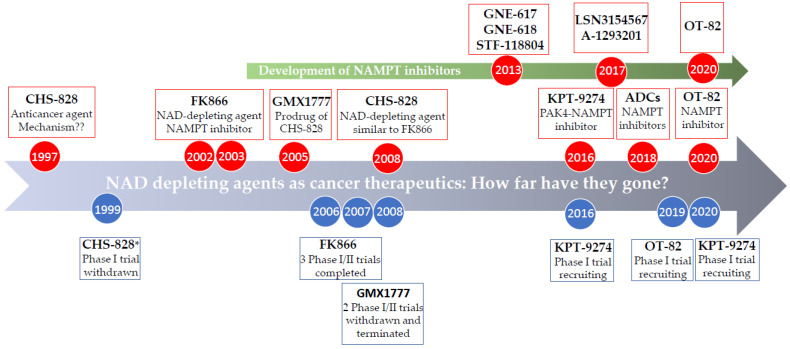
Timeline summary for the development of NAMPT inhibitors and their entry for evaluation in clinical studies. * the trial years are the years in which the study started according to clinicaltrials.gov.

**Figure 3 nutrients-13-01665-f003:**
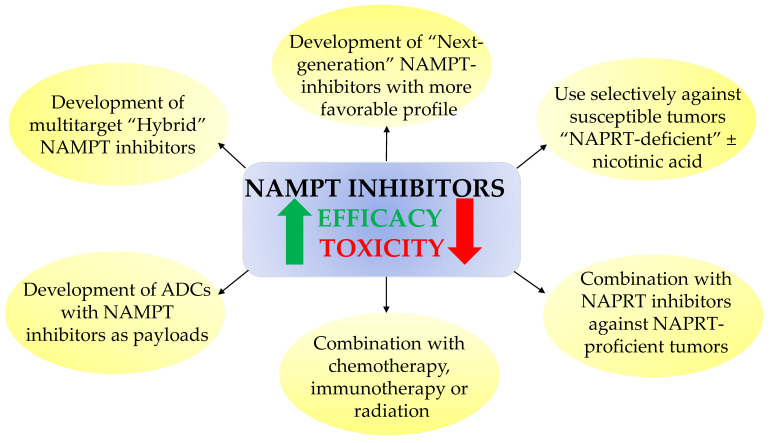
Schematic overview of the different strategies to improve the overall performance of NAMPT inhibitors.

**Table 1 nutrients-13-01665-t001:** Overview of the regulation of the major enzymes involved in NAD biosynthesis.

Regulator	Target	Mechanism	Effect	Cancer/Tissue Type
c-MYC and Max [37]	NAMPT	-Binding to and regulating the activity of the distal 4.6 kb putative *NAMPT* enhancer 65 kb downstream the *NAMPT* transcription start site specifically through the 1 kb “B-region” within the *NAMPT* enhancer.	Upregulation	Salvage-dependent cancer cells
c-MYC [38]	NAMPT	-Binding to the *NAMPT* promoter.	Upregulation	MCF-7 cells (breast cancer)
C/EBPβ [36]	NAMPT	-Interaction with *NAMPT* regulatory regions.	Upregulation	Mesenchymal GSCs
HMGA proteins [39]	NAMPT	-Binding to an *NAMPT* enhancer element during oncogene-induced senescence (OIS).	Upregulation	Oncogenic Ras-induced senescent IMR90 cells (lung fibroblasts)
SIRT6 [40]	NAMPT	-Regulation of NAMPT enzymatic activity through lysine deacetylation.	Upregulation	HEK293 cells (human embryonic kidney cells)
SIRT1 [41]	NAMPT	-Regulation of NAMPT activity through lysine deacetylation and secretion of eNAMPT.	Upregulation	Adipocytes
Foxo1 [42]	NAMPT	-Binding to conserved insulin response elements (IREs) in the *NAMPT 5′*-flanking promoter region.	Downregulation	MCF-7 cells (breast cancer)
NAMPT-AS “RP11-22N19.2” Lnc-RNA [43]	NAMPT	-Recruitment of the transcription factor POU2F2 to the promoter region of *NAMPT* to enhance NAMPT transcription. -Competitive binding to miR-548b-3p leading to increasing the *NAMPT* mRNA pool.	Upregulation	MDA-MB-231 and MDA-MB-468 cells (triple-negative breast cancer)
GACAT3 [44] Lnc-RNA	NAMPT	-Competitive binding to miR-135a, whose target gene is NAMPT.	Upregulation	U87 and U251 cells (glioma)
miR-381 [45]	NAMPT	-Post-transcriptional binding to the 3′- untranslated region (UTR) of NAMPT.	Downregulation	MDA-MB-231 and MCF-7 cells (breast cancer)
miR-206 [46]	NAMPT	-Binding to the 3′-UTR of NAMPT.	Downregulation	MDA-MB-231 and MCF-7 cells (breast cancer)
miR-494 [47]	NAMPT	-Binding to the 3’-UTR of NAMPT.	Downregulation	MDA-MB-231 and MCF-7 cells (breast cancer)
miR-154 [48]	NAMPT	-Binding to the 3’-UTR of NAMPT.	Downregulation	MDA-MB-231 and MCF-7 cells (breast cancer)
miR-26b [49]	NAMPT	-Binding to the 3′-UTR of NAMPT.	Downregulation	SW480 cells (colorectal cancer)
miR-206 [50]	NAMPT	-Regulation of NAMPT expression most probably through targeting the 3′-UTR of NAMPT.	Downregulation	MiaPaCa-2 and Panc-1 cells (pancreatic cancer)
miR-23b [51]	NAMPT	-Regulation of NAMPT expression.	Downregulation	melanoma cells
Gene Amplification [37]	NAPRT/ NADSYN	-Regulation of NAPRT or NADSYN expression.	Upregulation	PH-dependent tumors and cancer cell lines
Gene Silencing [52]	NAPRT	-Hypermethylation of *NAPRT* promoter region.	Downregulation	Several cancer cell lines
Mutant IDH1 [53]	NAPRT	-Hypermethylation of the CpG islands in the *NAPRT* promoter region and thus reprogramming NAD metabolism. -*IDH1*-mutant cancers are uniquely sensitive to NAMPT inhibitors via NAD depletion.	Downregulation	*IDH1*-mutant cancer cells (MGG119, MGG152, BT142, BT142, HT1080, 30T, SW1353)
Mutant PPM1D [54]	NAPRT	-Hypermethylation of the CpG islands in the genome and epigenetic silencing of *NAPRT* gene. -*PPM1D* mutant cancer cells are uniquely sensitive to NAMPT inhibitors.	Downregulation	*PPM1D* mutant astrocytes and diffuse intrinsic pontine glioma (DIPG) cell lines
SIRT3 [55]	NMNAT2	-Regulation of NMNAT2 activity through deacetylation.	Upregulation	A549 cells (non-small cell lung cancer)
miR-654-3p [56]	QAPRT	-Binding to the 3′-UTR of QAPRT.	Downregulation	Igrov-1 cells (ovarian cancer)
DSCAM-AS1 [57] Lnc-RNA	QAPRT	-Competitive binding of miRNA-150-5p and miRNA-2467-3p.	Upregulation	T47D and MCF-7 cells (breast cancer)
WT1 [58]	QAPRT	-Binding to a conserved site on the *QAPRT* promoter.	Upregulation	K562 cells (leukemia)

**Table 2 nutrients-13-01665-t002:** Summary of the reported combinations of NAMPT inhibitors with other agents in cancer therapy.

NAMPT Inhibitor	The Combination Agent/Drug	Class	Approval as an Anti-Cancer	Cancer Type	In Vitro Efficacy	In Vivo Efficacy	Ref.
FK866	5-fluorouracil	Antimetabolite	Yes	Gastric cancer	Yes	n/a ^1^	[71]
FK866	Fludarabine	Antimetabolite	Yes	Leukemia (CLL)	Yes	n/a	[135]
FK866	Etoposide	Topoisomerase II inhibitor	Yes	Leukemia	Yes	n/a	[136,137]
GMX1777	Etoposide	Topoisomerase II inhibitor	Yes	Lung cancer	n/a	Yes	[108]
FK866	Etoposide	Topoisomerase II inhibitor	Yes	Neuroblastoma	Yes	n/a	[138]
FK866	Cisplatin	Alkylating agent	Yes	Neuroblastoma	Yes	n/a	[138]
FK866 GMX1778	Cisplatin	Alkylating agent	Yes	Ovarian cancer	Yes	Yes (FK866)	[139]
FK866	Cyclosporin A Verapamil	Pgp inhibitor Pgp inhibitor	No No	Leukemia Leukemia	Yes Yes	n/a n/a	[140]
FK866	Bortezomib	Proteasome inhibitor	Yes	Multiple myeloma	Yes	Yes	[141]
FK866	Ibrutinib	Bruton’s tyrosine kinase inhibitor	Yes	Waldenstrom macroglobulinemia	Yes	Yes	[142]
GMX1777	Pemetrexed	Antimetabolite (Antifolate)	Yes	Non-small-cell lung cancer (NSCLC)	Yes	Yes	[143]
FK866	Gemcitabine	Antimetabolite	Yes	Pancreatic cancer (PDAC)	Yes	n/a	[144]
FK866	Gemcitabine	Antimetabolite	Yes	PDAC	Yes	Yes	[50]
STF-118804	Gemcitabine	Antimetabolite Topoisomerase II inhibitor Antimicrotubular agent	Yes	PDAC	Yes	n/a	[145]
	Etoposide		Yes	PDAC	Yes	n/a	
	Paclitaxel		Yes	PDAC	Yes	n/a	
FK866	Vorinostat Valproic acid	HDAC inhibitor HDAC inhibitor	Yes No	Leukemia Leukemia	Yes Yes	n/a n/a	[146]
GMX1778	^177^Lu-DOTATATE	Radiolabeled somatostatin analog	Yes	Neuroendocrine tumors	n/a	Yes	[147]
FK866	Rituximab	Anti-CD20	Yes	Lymphoma	Yes	Yes	[148]
FK866 GMX1778	Temozolomide	Alkylating agent	Yes	Glioblastoma	Yes	n/a	[149]
FK866 GMX1778	Temozolomide	Alkylating agent	Yes	*IDH1*-mutant cancers	Yes	Yes (FK866)	[150]
FK866	Olaparib	PARP inhibitor	Yes	Triple-negative breast cancer (TNBC)	Yes	Yes	[151]
GNE-618 FK866 GMX1778	Niraparib	PARP inhibitor	Yes	Ewing sarcoma	Yes	Yes (GNE-618)	[152]
OT-82	Niraparib	PARP Inhibitor	Yes	Ewing sarcoma	Yes	Yes	[153]
OT-82	Irinotecan & its metabolite SN-38	topoisomerase I inhibitors	Yes (Irinotecan)	Ewing sarcoma	Yes (SN-38)	Yes (Irinotecan)	[153]
OT-82	Cytarabine	Antimetabolite	Yes	Acute lymphoblastic leukemia (ALL)	Yes	Yes	[154]
OT-82	Dasatinib	Tyrosine kinase inhibitor	Yes	ALL	n/a	Yes	[154]
OT-82	Etoposide	Topoisomerase II inhibitor	Yes	ALL	Yes	n/a	[154]
GMX1778	Anti-mouse PD-1 antibody	Immune checkpoint inhibitor	Human anti-PD1: Yes	Glioblastoma	n/a	Yes	[155]
MV87	Anti-mouse PD-1 antibody	Immune checkpoint inhibitor	Human anti-PD1: Yes	Fibrosarcoma	n/a	Yes	[156]
KPT-9274	Anti-mouse PD-1 antibody	Immune checkpoint inhibitor	Human anti-PD1: Yes	Renal cell carcinoma	n/a	Yes	[157]
KPT-9274	Anti-mouse PD-1 antibody	Immune checkpoint inhibitor	Human anti-PD1: Yes	Melanoma Colon adenocarcinoma	n/a	Yes (PAK4)	[158]
KPT-9274	Bendamustine Melphalan	Alkylating agent Alkylating agent	Yes Yes	Waldenstrom macroglobulinemia	Yes Yes	Yes n/a	[159]
KPT-9274	Everolimus	mTOR inhibitor	Yes	Pancreatic neuroendocrine tumor	Yes	n/a	[160]
KPT-9274	Gemcitabine Oxaliplatin	Antimetabolite Alkylating agent	Yes Yes	PDAC PDAC	Yes Yes	Yes (PAK4) n/a	[161]
FK866	TRAIL	Apoptosis activator	Not approved as a drug	Leukemia	Yes	n/a	[162]
FK866	2-HNA	NAPRT inhibitor	Not approved as a drug	Ovarian cancer Pancreatic cancer	Yes	Yes (sodium salt of 2-HNA)	[62]
FK866	L-1-methyl-tryptophan	Indoleamine 2,3-dioxygenase (IDO) inhibitor	Not approved as a drug	Gastric cancer Bladder cancer	n/a	Yes (only in immuno-competent mice)	[163]
FK866	β-Lapachone	ROS generator & NQO1 substrate	Not approved as a drug	PDAC NSCLC	Yes	n/a	[164,165,166]
FK866	FX11	Lactate dehydrogenase A (LDHA) inhibitor	Not approved as a drug	Lymphoma	Yes	Yes	[167]
FK866	1-methyl-3-nitro-1-nitrosoguanidinium (MNNG)	Alkylating agent	Not approved as a drug	Leukemia	Yes	n/a	[168]

^1^ n/a: not available (i.e., not reported in the referred article).

**Table 3 nutrients-13-01665-t003:** Summary of the reported preclinical studies of the NAMPT inhibitors that are currently being evaluated in clinical trials.

Compound	Cancer Type	Cancer Cell Lines	In Vitro Effects	Mouse Model	In Vivo Model	In Vivo Effects	Reported Mode of Action
KPT-9274	Renal cell carcinoma (RCC) [118]	RCC cell lines: 786-O ACHN Caki-1	-Attenuation of viability, invasion, and migration in several RCC cell lines. -Limited toxicity in normal human renal epithelial cells. -Induction of apoptosis. -Decrease in G2-M transition. -Reduced NAD and SIRT1 levels. -NA rescued NAD levels in normal renal epithelial cells but not in 786-O and Caki-1 “NAPRT deficient” cells. -Reduction in nuclear β-catenin and of the Wnt/β-catenin targets c-MYC and cyclin D1 as a result of PAK4 inhibition.	Male nude mice	RCC xenograft model: 786-O cells	-Reduced tumor growth. -No significant animal weight loss.	PAK4 and NAMPT inhibition
KPT-9274	Renal cell carcinoma (RCC) [157]		Ex vivo: -Reduced tumor expression of PAK4 and phospho-β-catenin. -NAD + NADH levels in tumors decreased by KPT-9274 and increased by anti-PD1 antibody.	Male BALB/cJ mice	RCC allograft model: Mouse RENCA-luciferase (RENCA-Luc) cells	-Significant reduction in tumor growth with KPT-9274 and anti-PD1 combination compared to each agent alone. -No significant animal weight loss.	PAK4 and NAMPT inhibition
KPT-9274	Pancreatic ductal adenocarcinoma (PDAC) [161]	PDAC cell lines: MiaPaCa-2 HPAC Panc1 Colo-357 L3.6pl MiaPaCa-2 cancer stem cells	-Inhibition of proliferation of PDAC cells. -Limited toxicity in normal pancreatic human epithelial cells. -Cancer-selective induction of apoptosis and cell-cycle arrest. -Suppression of cancer migration. -Overcoming stemness (PDAC cancer stem cells) and downregulation of EMT markers. -Synergistic effect with gemcitabine and oxaliplatin.	Female SCID mice	PDAC xenograft model: L3.6pl cells AsPc-1 cells PDAC cancer stem cell xenograft model: CD44^+^/CD133^+^/EpCAM^+^ MiaPaCa-2 cells	-Remarkable antitumor activity as a single agent. -Marginal antitumor activity in combination with gemcitabine. -No signs of toxicity. -Suppression of growth of highly resistant PDAC cancer stem cell-derived tumors.	PAK4 inhibition
KPT-9274	Acute myeloid leukemia (AML) [169]	AML cell lines: HL-60 THP-1 Kasumi-1 MV4-11 OCI-AML3 MOLM13 Primary AML cells	-Inhibition of proliferation of AML cells. -Cell cycle arrest. -Induction of apoptosis. -Reduction in NAD levels, disruption of mitochondrial activity, and cellular respiration. -Limited toxicity on normal hematopoietic cells.	NSG mice	AML xenograft model: luciferase-positive MV4-11 cells Patient-derived xenograft (PDX) model of AML	-Improved overall survival. -Reduced disease progression and tumor burden.	NAMPT inhibition
KPT-9274	B-cell acute lymphoblastic leukemia (B-ALL) [194]	B-ALL cell lines: KOPN-8 RS4;11 REH 697 OP-1 Nalm6 Sup-B15 SEM PDX B-ALL cells	-Strong inhibition of cell growth. -Induction of apoptosis. -Intracellular NAD depletion and modulation of NAD-dependent pathways. -NA supplementation reversed KPT-9274-mediated growth inhibition in three sensitive B-ALL cell lines.	NSG mice	PDX model of B-ALL: luciferase-transduced LAX2 cells	-Effective suppression of leukemia progression. -Significantly improved survival. -Acceptable adverse effect profile (normal mice activity, no significant difference in body weight between groups).	NAMPT inhibition
KPT-9274	Triple-negative breast cancer (TNBC) [195]	BC cell lines: MDA-MB-231 MDA-MB-468 SUM159 MCF7 SkBr-3 BT-474	-Inhibition of cell proliferation in several BC cell lines. -Reduction in viability was more pronounced in TNBC cell lines (almost complete inhibition). -Stimulation of apoptosis.	Female nude mice	TNBC xenograft models: MDA-MB-231 cells MDA-MB-468 cells SUM159 cells	-Significant reduction in tumor weights and volumes. -No significant effect on the body weights of the mice. -Reduced PAK4 protein levels in tumors.	PAK4 inhibition
KPT-9274	Melanoma [158]	Melanoma cell lines: Murine B16 cells		C57BL/6 mice	Melanoma model: B16 cells	-Significant decrease in tumor growth with KPT-9274 and anti-PD1 combination compared to each agent alone.	PAK4 inhibition
KPT-9274	Colon cancer [158]	Colon cancer cell lines: Murine MC38 cells		C57BL/6 mice	Colon adenocarcinoma model: MC38 cells	-Significant decrease in tumor growth with KPT-9274 alone or combined with anti-PD1 compared to anti-PD1 alone.	PAK4 inhibition
KPT-9274	Pancreatic neuro-endocrine tumors (PNET) [160]	PNET cell lines: BON-1 QGP-1	-Reduction in growth and survival of PNET cells. -Reduced NAD and ATP levels and ATP collapse was reversed by NA. -Synergistic effect with everolimus.	Female SCID mice	PNET xenograft model: BON-1 cells	-Significant reduction in tumor growth as monotherapy.	PAK4 and NAMPT inhibition
KPT-9274	Waldenstrom macroglobulinemia (WM) [159]	WM cell lines: BCWM-1 MWCL-1 RPCIWM-1 Primary WM cells	-Reduction in cell viability. -NA rescued BCWM-1 cells from KPT-9274-mediated growth inhibition. -Impairment of DNA repair and induction of DNA damage. -Induction of apoptosis. -Synergistic effect in combination with DNA-damaging agents (bendamustine & melphalan).	SCID mice	WM xenograft model: BCMW-1 cells	-Significant inhibition of tumor growth as a single agent. -Significant reduction in tumor volume with KPT-9274 and bendamustine combination compared with either agent alone.	PAK4 and NAMPT inhibition
KPT-9274	Multiple myeloma (MM) [198]	Many human myeloma cell lines Primary MM cells	-Reduction in cell growth and survival in a large panel of MM cell lines and primary MM cells. -Suppression of the promoting effects of the bone marrow microenvironment. -No significant effect on bone marrow mononuclear cells or PBMCS. -Induction of apoptosis and deregulation of the MEK/ERK pathway.	Nude mice	MM xenograft models: MM1S cells OPM2 cells	-Significant single-agent antitumor activity in both MM xenograft models. -Higher sensitivity to KPT-9274 was seen with t(4:14) FGFR3-mutated OPM2 tumors compared to MM1S xenografts.	PAK4 inhibition
KPT-9274	Ewing sarcoma (EWS) [197]	EWS cell lines: CHLA-10 A673 TC32	-Suppression in cell proliferation. -Reduction in invasive and migratory characteristics. -Potential synergistic effect with doxorubicin and vincristine.	NSG mice	EWS xenograft models: A673 cells TC71 cells PDX and metastatic models of EWS	-Significant reduction in tumor growth as a single agent. -Reduced metastatic burden in the EWS metastatic model.	PAK4 inhibition
KPT-9274	Rhabdomyosarcoma (RMS) [196]	RMS cell lines: RH30 RD RH4	-Reduction in cell proliferation in multiple RMS cell lines (IC_50_ ranges from 40 to 80 nM). -Limited toxicity in normal skeletal muscle myoblast cells (IC_50_ values were 8–10 times higher). -Induction of apoptosis and G1-S arrest. -Inhibition of cell motility and invasive properties. -Reduced PAK4 activity and β-catenin activation.	NSG mice	RMS orthotopic xenograft models: RH30 cells RD cells PDX model of relapsed RMS Metastatic model of RMS: RD cells	-Significant reduction in tumor growth in the orthotopic and PDX models with KPT-9274 alone compared to vehicle. -No significant changes in body weight. -Reduced metastatic burden in the liver with KPT-9274 in the metastatic model.	PAK4 inhibition
OT-82	Pediatric acute lymphoblastic leukemia (ALL) [154]	ALL cell lines: PER-826A REH PER-703A CEM PER-485 RS4;11 KOPN-8 Loucy Pediatric PDX ALL cells	-Potent dose-dependent reduction in cell viability. -Reduction in NAD levels, ATP levels, and PARP activity. -Higher DNA damage. -Induction of apoptosis. -Synergistic effect with cytarabine and etoposide in PER-485 cell line.	NOD/SCID mice NSG mice	21 PDX models of high-risk pediatric ALL	-Significant extension of the survival in 20/21 (95%) PDXs and objective response in 18/21 (86%) PDXs→ significant leukemia regression. -Therapeutic enhancement when combined with cytarabine in 2 aggressive MLLr-ALL PDX models and with dasatinib in one Ph+ ALL PDX model.	
OT-82	Ewing sarcoma (EWS) [153]	EWS cell lines: TC71 TC32 RDES SK-N-MC EW8 5838 CHLA-258	-Potent inhibition in cell growth and proliferation. -Reduction in NAD levels and PARP activity. -Higher DNA damage. -G2 arrest and induction of apoptosis. -Enhanced antiproliferative effects when combined with niraparib or SN-38.	SCID beige mice NOG mice	Orthotopic xenograft models of EWS: TC71 cells TC32 cells PDX model of EWS	-Significant reduction in tumor growth and prolongation of survival with doses of 25 mg/kg and 50 mg/kg. -Cessation of treatment resulted in tumor regrowth. -Improved OT-82 efficacy when combined with irinotecan or niraparib (slower tumor growth and prolongation of median survival). -Several unexpected deaths occurred with OT-82/niraparib combination.	
OT-82	Acute myeloid leukemia (AML) [109]	AML cell lines: MV4-11 THP1	-Potent reduction in cell viability. -Depletion of NAD and ATP levels. -Induction of apoptosis. -NA addition rescued MV4-11 cells from OT-82-mediated cytotoxicity.	SCID mice	Subcutaneous (SC) and systemic xenograft models of AML: MV4-11 cells	-Significant dose-dependent reduction in tumor volume→ SC model. -Significant prolongation of mice survival at 25 and 40 mg/kg → systemic model. -Improved survival with the optimized OT-82 regimen→ systemic model.	
OT-82	Erythroleukemia [109]	Erythroleukemia cell line: HEL92.1.7	-Potent reduction in cell viability.	SCID mice	SC and systemic xenograft models of erythroleukemia: HEL92.1.7 cells	-Significant reduction in tumor volume at 50 mg/kg dose→ SC model. -Significant prolongation of mice survival at 40 mg/kg→ systemic model.	
OT-82	Burkitt lymphoma (BL) [109]	BL cell lines: Raji Ramos	-Potent reduction in cell viability.	SCID mice	SC xenograft model of BL: Ramos cells	-Significant reduction in tumor volume with the optimized OT-82 regimen.	
OT-82	Multiple myeloma (MM) [109]			SCID mice	SC xenograft model of MM: RPMI 8226 cells	-Significant reduction in tumor volume with the optimized OT-82 regimen.	

## Data Availability

Not applicable.

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
