# Peer review of "Advances in NAD-Lowering Agents for Cancer Treatment"

_nutrients, 2021, doi:10.3390/nu13051665_

Round 1
Reviewer 1 Report
This is an outstanding review on the role of NAD metabolism as therapeutic target in cancer. The review is well-written and provides a timely and fairly comprehensive overview of this hot topic of translational cancer research. The authors are renowned experts in their field, the article is well structured and will thus likely be highly valuable to readers interested in NAD metabolism in cancer cells and will hopefully spur further studies in this area of research.
Therefore, I wholeheartedly recommend publication of this interesting manuscript in its current form.
Author Response
Attached please find our revised version of the article “Advances in NAD-Lowering Agents for Cancer Treatment”

Reviewer 2 Report
This is a timely, extremely comprehensive and particularly well articulated review on the complex topic of approaching NAD depletion in the context of cancer.
The article presents a very informative review of the current literature in the field in the broadest context of cancer therapies.
The only minor point that might would be worthwhile raising is that the NAMPT/NRK axis vs NAPRT/NADSYN axis of NAD maintenance are not as simple as the current knowledge leads to believe. For NRK to appropriately contribute to the NAD biosynthetic challenge, NR must be available and the only possible source of NR, beside NAD from NUDIX cleavage (which would be counter-productive), is the reduced form of NR, NRH. In ddition to the large body of work on NRQ/NQO2, the use of paracetamol, an NQO2 substrate and an NRH to NR enhancer, could have a valuable contribution to make to the idea. NRH is the degradation product of NADH by NUDIXs like NUDIX5, 12, and 13. Surely, the role of NADH in the maintenance of NAD should not be overlooked and should be put in the context of IDH2 mutants, as well as NQO1/Lapachone/NAMPT combination. Processes that enhance NADH to NAD conversion, in the context of an NAD consuming cell lines will enhance the NAMPT inhibitors' efficacy, and again could have been introduced in this review and would be in the scope of this review.
Further, the authors have brought forth the role of the microbiome to the NA production. However this point should have been followed on by the fact that gut NA is rapidly converted to Nam by the liver, and that circulating NA has very low abundance. Therefore, the mechanisms by which NAPRT-dependent cancer cells remain viable upon NAMPT-inhibition should be looked from a different angle in terms of sources of substrate than that of NAMPT-dependent cells. Nam is recycled to Nam; NA is converted to Nam in NAPRT-dependent tumor cells. Where does the bioavailability of the NAPRT substrate come from?
Finally, the authors have made no mention of the role of NNMT in the regulation of Nam bioavailability as well as the hypomethylation of genes encoding for NAD biosythetic enzymes in cell culture and in tumors. This factor, while somewhat peripheral is nonetheless central to be discussion: what happens to NAD and its precursors in the context of cancer, and how can this be best harnessed if one focuses on what is perceived as the most efficient substrate to NAD biosynthesis.
Finally, the authors have completely overlooked the PRPP pathway, its role in the production of the riboside building block of NAD and its link to glucose transporters and NADPH production, etc... Cells with a need for NADPH for ROS maintenance could rely more on the PRPP pathway offering increased concentration of nucleotide building blocks, including AICAR/AMP and NMN. This side of the literature should at least be mentioned. In some instance, the review positions this biology in the context of Nam while it should be in the context of the riboside substrate and NAD sugar moieties.
Author Response

(The authors gave the same response as above.)
